# Deoxynivalenol Modulates the Viability, ROS Production and Apoptosis in Prostate Cancer Cells

**DOI:** 10.3390/toxins11050265

**Published:** 2019-05-11

**Authors:** Dominika Ewa Habrowska-Górczyńska, Karolina Kowalska, Kinga Anna Urbanek, Kamila Domińska, Agata Sakowicz, Agnieszka Wanda Piastowska-Ciesielska

**Affiliations:** 1Laboratory of Cell Cultures and Genomic Analysis, Department of Comparative Endocrinology, Medical University of Lodz, Zeligowskiego 7/9, 90-752 Lodz, Poland; dominika.habrowska@umed.lodz.pl (D.E.H.-G.); karolina.kowalska1@umed.lodz.pl (K.K.); kinga.urbanek@umed.lodz.pl (K.A.U.); 2Department of Comparative Endocrinology, Medical University of Lodz, Zeligowskiego 7/9, 90-752 Lodz, Poland; kamila.dominska@umed.lodz.pl; 3Department of Medical Biotechnology, Medical University of Lodz, Zeligowskiego 7/9, 90-752 Lodz, Poland; agata.sakowicz@umed.lodz.pl

**Keywords:** mycotoxin, deoxynivalenol, prostate cancer, apoptosis, oxidative stress

## Abstract

Deoxynivalenol (DON), known as vomitoxin, a type B trichothecene, is produced by *Fusarium*. DON frequently contaminates cereal grains such as wheat, maize, oats, barley, rye, and rice. At the molecular level, it induces ribosomal stress, inflammation and apoptosis in eukaryotic cells. Our findings indicate that DON modulates the viability of prostate cancer (PCa) cells and that the response to a single high dose of DON is dependent on the androgen-sensitivity of cells. DON appears to increase reactive oxygen species (ROS) production in cells, induces DNA damage, and triggers apoptosis. The effects of DON application in PCa cells are influenced by the mitogen-activated protein kinase (MAPK) and NFΚB- HIF-1α signaling pathways. Our results indicate that p53 is a crucial factor in DON-associated apoptosis in PCa cells. Taken together, our findings show that a single exposure to high concentrations of DON (2–5 µM) modulates the progression of PCa.

## 1. Introduction

Deoxynivalenol (DON), or vomitoxin, is a type-B trichothecene mycotoxin produced by plant fungi from *Fusarium* species, and is reported to be one of the most prevalent mycotoxins worldwide. DON is commonly found in wheat, barley, and corn worldwide [1]. The prevalence of DON in food samples collected in the European Union has been found to be 57% [2]. Cereal grains are widely contaminated with DON, with 57% of wheat, 41% of maize, 68% of oats, 59% of barley, and 27% of rice believed to be affected. DON is only partially destroyed in food processing during the high temperature stage, and is thus present in processed cereals [3]. After consumption, DON is quickly metabolized and is no longer present in the serum after 24 h due to its quick absorption in the gastrointestinal tract and excretion in the urine, mainly in glucuronidated forms [4]. The temporary tolerable daily intake of DON established by the World Health Organization (WHO) is 1 µg kg^−1^ body weight. The maximum levels of DON in cereal products established by the European Commission in 2006 are 1.75 mg kg^−1^ for unprocessed durum, maize, and oats, 1.25 mg kg^−1^ for flour and much less for infant food (0.2 mg kg^−1^) [5]. The No Observed Adverse Effect Level (NOAEL) of DON was established to 0.04 mg kg^−1^ body weight, based on subacute and subchronic toxicity studies [6]. Interestingly, according to a European Food Safety Agency (EFSA) report, infants present the highest chronic dietary exposure to DON. Significant amounts of DON and its two major metabolites (3-acetyl-DON and 15-acetyl-DON) have also been reported in adolescents and adults in Europe, indicating a potential health concern [6].

Although a considerable body of animal studies has shown that DON is genotoxic, impairs the immune response, and exhibits both developmental and reproductive toxicity through the reduction of fertility, embryotoxicity, and postnatal mortality [4], accordingly to the newest report of the International Agency for Research on Cancer (IARC) DON is classified as *not classifiable as to its carcinogenicity to humans* (Group 3). This group of agents include both non-cancerogenic agents with documented toxicity as well as agents with no sufficient evidence to be determined as toxic, which trigger a different animal and human effect or indicating gaps in research studies [7]. Acute exposure to DON triggers diarrhea, vomiting, leukocytosis, and hemorrhaging [6]. On the molecular level, DON indirectly alters DNA and RNA synthesis by binding to ribosomes and directly altering protein synthesis. It is reported to disrupt mitochondria function, modulate cell membrane integrity and induce apoptosis in eukaryotic cells [8]. It has been found to be highly toxic against cultured primary rat hepatocytes [9,10], porcine hepatocytes [11], RAW 264.7 murine macrophages [12], human monocytes [13], human pre-T lymphocytes, pre-B lymphocytes, hamster kidney-derived BHK21 cells, mouse hepatoma cell line MH-22a [14], and Jurkat T-lymphocytes [15]. It also induces apoptosis in lymphoid organs [16,17] and modulates cell-mediated immunity in a dose-dependent manner [18]. It is reported to induce oxidative stress in cells by the production and accumulation of intracellular reactive oxygen species (ROS) and the induction of programmed cell death [19].

Oxidative stress disturbs cell homeostasis and viability, and induces a variety of cellular responses via the generation of ROS [20]. It has been suggested that the incidence of prostate cancer (PCa) is associated with excessive ROS production and a reduction in antioxidant activity. Moreover, PCa and benign prostatic hyperplasia (BPH) are also associated with oxidative stress [21]. Tumor cells are able to overestimate or inhibit the molecular pathways responsible for proliferation, survival, and programmed cell death [22]. In these cases, compounds that modulate the oxidative stress and antioxidant defense mechanisms in cells might be a crucial environmental factor in modulating the molecular events associated with PCa progression and metastases. Although DON is not considered a carcinogen for humans [6], its regulation of ROS production in tumor cells might indirectly assist the progression of tumors via the apoptosis process.

Therefore, the aim of the present study is to determine whether DON might induce oxidative stress and apoptosis in prostate cancer cells in non-chronic conditions (24 h exposure), mimicking acute exposure to DON (>1 µM). The androgen ratio and androgen receptor (AR) expression in PCa patients plays a crucial role, both in the process of carcinogenesis and in the progression of the tumor [23]. As DON is reported to modulate the process of steroidogenesis in animals through the modulation of testosterone [24], various androgen-dependent (LNCaP) and androgen-independent (DU-145, PC3) *in vitro* prostate cancer models were used to evaluate the DON-induced oxidative stress in PCa, as well as castration-resistant (22Rv1) models.

## 2. Results

### 2.1. DON Decreases Viability of Prostate Cancer Cells

To verify if DON, in a single exposure, modulates ROS production in PCa cells, all experiments were conducted after 24 h of exposure. The cell lines used represent different *in vitro* models of PCa. PC3, DU-145, and LNCaP cells are considered a gold standard for PCa cell lines. DU-145 is derived from a brain metastasis of PCa, does not express AR or prostate specific antigen (PSA) at the mRNA or protein level and is considered androgen-independent, similar to PC3 derived from vertebral metastatic prostate tumors. PC3 cells also express aberrant p53 and are PTEN deficient. LNCaP cells derived from a lymph node metastasis of PCa are androgen-dependent and display AR and PSA expression. 22Rv1 cells were isolated from a xenograft and are positive for AR and PSA only at the mRNA level. The tested doses of DON (0.0001–40 µM) were based on previous studies conducted on cancer cells [25,26].

Following DON exposure, cell viability, measured with an MTT assay, fell in a concentration-dependent manner (Figure 1A–D). The cell viability was significantly decreased after DON exposure below 0.3 µM for 22Rv1 cells, below 2 µM for LNCaP, PC3 and DU-145 cell (**** p* < 0.001). Concentrations of DON which decreased cell viability to 60–70% of control values (2–5 µM DON) were used in the remaining experiments. The highest tested dose of DON (40 µM) caused about a 30% (22Rv1, LNCaP), 50% (PC3), and 60% (DU-145) decrease in cell viability as compared to controls, indicating that different PCa cell lines differ in their sensitivity. Other differences in sensitivity were also visible with regard to cell morphology (Figure 1E). The morphology of 22Rv1 and LNCaP cells was only slightly changed and PC3 cells were not visibly dividing, but dead cells were visible in the DU-145 cells treated with 5 µM of DON.

### 2.2. DON Induces ROS Production in PCa Cells

The next stage examined whether DON induces oxidative stress in PCa cells. The percentage of cells with increased intracellular oxidative activity (ROS positive cells) was calculated, and the expression of major antioxidant enzymes was evaluated: superoxide dismutase 1 (SOD1), superoxide dismutase 2 (SOD2), and the activity of enzymes SOD2 and glutathione peroxidase (GPx). ROS production was measured using the Oxidative Stress Muse^®^ Kit, which uses dihydroethidium to detect ROS in cellular populations. DON was found to cause a statistically significant increase in ROS production in all tested cell lines (**** p* < 0.001) with the exception of PC3 cells, for which only an insignificant rise was observed (Figure 2A–D). The PC3 cells displayed a different tendency regarding SOD2 activity. For all cell lines, with the exception of PC3 and the 5 µM dose in DU-145 cells, a decreasing, concentration-dependent tendency in SOD2 activity was observed (Figure 2G–J). No significant GPx activity was observed at the concentration of NADPH (Figure 2K–N). SOD1 and SOD2 expression was also evaluated at the mRNA and protein levels: DON caused a statistically significant increase (** *p* < 0.01, *** *p* < 0.001) in the expression of SOD1 in the androgen-responsive cell lines LNCaP and 22Rv1, but a statistically significant decrease in the androgen-independent cell lines PC3 and DU-145 (* *p* < 0.05, *** *p* < 0.001) (Table 1). A similar effect was observed at the protein level (representative results, Figure 2F). The expression of *SOD2* was significantly decreased by DON in DU-145 cells (**** p* < 0.001) and increased in PC3 and LNCaP cells (** p* < 0.05, ** *p* < 0.01, *** *p* < 0.001) (Table 1). No significant changes in the expression of *SOD2* were observed in 22Rv1 cells, although a decreasing tendency was present. Similar changes in expression at the protein level were observed (Figure 2F).

DON was previously reported to modulate MAPK signaling [27] and influence the oxidative response genes [28]. The NFKB signaling pathway is one of the main pathways involved in the response to oxidative stress. NFKB is also reported to modulate HIF-1α expression, the main transcription factor activated by low oxygen concentrations [29]. Another transcription factor, FOXO3a, plays a crucial role in DON toxicity, both through its documented role in the regulation of steroidogenesis as well as its regulation of apoptosis [30]. Accordingly, we evaluated the expression of *RelA*, *FOXO3A,* and *HIF-1α* as well as ERK and JNK. DON caused a statistically significant increase in the expression of *RelA*, *FOXO3A,* and *HIF-1α* (**** p* < 0.001) as compared with controls (Table 2). The expression of RelA was also evaluated at the protein level, and similar results were observed (representative Western blots are presented in Figure 3). The increase in the protein expression of the phosphorylated form of ERK (p-ERK) was observed in 22Rv1, PC3, and DU-145 cell lines, whereas the expression of the phosphorylated form of JNK (p-JNK) was elevated in LNCaP, PC3, and DU-145 cells, indicating that DON might modulate the expression of MAPK signaling pathway proteins (Figure 3).

### 2.3. DON Induces Apoptosis in PCa Cells

Oxidative stress triggers apoptosis in tumor cells [31]. Due to this fact, the next part of the experiment evaluated whether DON induces apoptosis in PCa cells. Firstly, apoptotic cells were stained with Annexin V (Muse Annexin V Dead Cell Kit) and counted. DON was found to cause a statistically significant increase in the number of early and late apoptotic cells in 22Rv1, LNCaP, and DU-145 cell lines (*** p* < 0.01, *** *p* < 0.001) (Figure 4A–E). Interestingly, a significant decrease in the number of total apoptotic cells was observed for PC3 cells treated with DON (**** p* < 0.001) (Figure 4C); although no significant increase in ROS generation was observed, cell viability decreased. The DU-145 cell line demonstrated both the greatest increase in the number of apoptotic cells and the greatest decrease in cell viability observed in the MTT assay.

The next part of the study evaluated the expression of apoptosis-associated genes—namely caspase 3 (*Casp3*) and Associated X, Apoptosis Regulator (*Bax*)—using RT-qPCR. DON application caused a statistically significant increase in *Casp3* expression (** p* < 0.05, ** *p* < 0.01, *** *p* < 0.001) in all lines apart from DU-145 (Table 3). *Bax* expression was significantly increased after DON exposure for the 22Rv1, LNCaP, and DU-145 cell lines. A significant decrease in the expression of *Bax* was observed in PC3 cells (** p* < 0.05). Following this, the expression of apoptotic proteins phospho-p53, p53, cleaved Casp3 and Poly (ADP-ribose) polymerase (PARP) was evaluated (Figure 5). A statistically significant increase in p53 expression was observed in 22Rv1 and DU-145 cells (**** p* < 0.001), but the contrary was observed in the LNCaP and PC3 cell lines (*** p* < 0.01, *** *p* < 0.001). In all cell lines with the exception of PC3, the expression of the phosphorylated form of p53 was significantly increased after DON treatment (**** p* < 0.001). The expression of the other two markers of apoptosis, cleaved Casp3 and PARP, was significantly increased in all cell lines except PC3 (*** p* < 0.01, *** *p* < 0.001), which correlates with the presence of apoptotic cells, as described previously.

### 2.4. DON Modulates the Mitochondrial Potential in PCa Cells

Apoptosis is typically initiated as a consequence of oxidative stress, DNA damage or mitochondrial dysfunction [32]. Induction following DON application is associated with changes in the mitochondrial membrane potential and then the release of cytochrome c. The effect of DON on the mitochondrial membrane potential in the 22Rv1, LNCaP, PC3, and DU-145 cells was investigated with the Muse^®^ MitoPotential Kit. Cells with depolarized mitochondria exhibit a decrease in the intensity of fluorescence. A dead cell marker (7-AAD) was also used as an indicator of the integrity of the cell membrane and as a marker for cell death. DON induced depolarization of mitochondria, demonstrated by a statistically significant decrease (**** p* < 0.001) in fluorescence without a decrease in the cell membrane structural integrity, as seen in Figure 6, which indicates that the mitochondrial membrane potential of the 22Rv1, LNCaP, and PC3 cells collapsed following DON treatment, but the cell membrane structural integrity was intact. In DU-145 cells, DON induced a significant depolarization of the mitochondria (**** p* < 0.001) and disrupted the structural integrity of the cell membrane, statistically significant for all doses of DON (**** p* < 0.001).

### 2.5. DON Induces DNA Damage in PCa Cells

The induction of apoptosis is also associated with the presence of DNA damage. As increased PARP expression was observed, indicative of DNA repair, the next part of the study examined whether DON induces DNA damage in 22Rv1, LNCaP, PC3, and DU-145 cells. The activation of ataxia-telangiectasia mutated protein kinase (ATM) and H2A histone family member X (H2A.X) was investigated with the Muse^®^ Multi-Color DNA Damage Kit. The results showed an increase in the percentage of cells with damaged DNA in all tested cell lines (Figure 7). The increase in DNA damage in 22Rv1 cells was statistically significant for 5 µM and 2 µM DON doses (*** p* < 0.01, ** p* < 0.05), but the LNCaP cells experienced statistically significant DNA damage only for the 5 µM DON dose (**** p* < 0.001). In the PC3 and DU-145 cells, statistically significant increases in the number of cells with damaged DNA were observed for all tested doses (**** p* < 0.001 or *** p* < 0.01 as indicated in Figure 7). The presence of DNA damage was also visible in DAPI staining as fragmentated nuclei (Figure 7E).

### 2.6. DON Modulates Cell Cycle in PCa Cells

Next, the effect of DON on the cell cycle progression of the four investigated PCa cell lines was evaluated. Muse^®^ Cell Cycle analysis was performed on cells treated with DON (2–5 µM) for 24 h. The results showed a decrease in the number of cells in the G0/G1 cell cycle phase as compared to the control cells (Figure 8). The decrease was dependent on the concentration of DON for all tested cell lines. A statistically significant decrease in the percentage of 22Rv1 cells was observed for 5 µM DON treatment (*** p* < 0.01). In LNCaP and DU-145 cells, a statistically significant dose-dependent decrease was noticed (**** p* < 0.001). A decrease was also observed in PC3 cells, although it was statistically significant only for the 5 µM (**** p* < 0.001) and 2 µM (** p* < 0.05) doses. The decrease in the number of cells in the G0/G1 cell cycle phase might be associated with the increased expression of cyclin D1 and cyclin-dependent kinase 4 (*CCND1* and *CDK4*), whose expression was significantly increased for all cell lines (Table 4).

The treatment with DON caused an increase in the percentage of cells in the S phase of the cell cycle for all tested lines. The statistically significant increase in the percentage of 22Rv1 cells in the S cell cycle phase was observed for all tested doses (*** p* < 0.01, *** *p* < 0.001) of DON. The increase in the S cell cycle phase was also observed in PC3 and DU-145 cells, although it was statistically significant only for the 5 µM DON dose (**** p* < 0.001) in both lines and for the 3 µM DON dose (* *p* < 0.05) for the PC3 cells.

DON treatment was found to increase significantly the number of 22Rv1 cells in the G2/M cell cycle phase for the 3 µM and 2 µM doses (**** p* < 0.001). The increase in the percentage of cells in the G2/M cell cycle phase in the LNCaP model was statistically significant for all tested doses (**** p* < 0.001). A similar and dose-dependent increase in the number of cells was observed in the PC3 and DU-145 cells, although it was statistically significant only for the PC3 cells (*** p* < 0.01, *** *p* < 0.001). For the 22Rv1, LNCaP, and DU-145 cells, we observed the presence of the cells in the subG0 cell cycle phase, as indicated by the process of apoptosis. The increase in the percentage of cells in the subG0 cell cycle phase in the 22Rv1 and LNCaP cells was dose-dependent and statistically significant at all tested doses (*** p* < 0.01, *** *p* < 0.001). Similarly, in DU-145 cells, an increase in the percentage of cells in the subG0 cell cycle phase was observed and statistically significant for 5 µM (**** p* < 0.001) and 2 µM (** p* < 0.05) doses of DON. The results are similar to the previously-observed increase in the number of apoptotic cells of the 22Rv1, LNCaP, and DU-145 cell lines after DON exposure. No increases in the number of apoptotic cells or the number of cells in the subG0 cell cycle phase were observed in the PC3 cells.

The cyclin B1 and cyclin-dependent kinase 1 (*CCNB1* and *CDC2*) complex is a main regulator of the G2/M cell cycle progression. We observed that the expression of *CCNB1* and *CDC2* was significantly increased after DON treatment in the LNCaP and DU-145 cells. A similar but not statistically significant increase was observed for the 22Rv1 cell line. Interestingly, the expression of genes was significantly decreased in the PC3 cells, although an increase in the number of cells in the G2/M cell cycle phase was observed. We also evaluated the expression of the cyclin-dependent kinase inhibitor p21 (*CDKN1A*), and similar to previous results, we observed a significant increase in its expression for all cell lines, with the exception of the PC3 cells. p21 is known to be controlled by p53 and is responsible for the execution of apoptosis after Casp3 action. The lack of the subG0 phase in PC3 cells, as well as the lack of changes in the expression of p21, is associated with the absence of the p53 protein.

## 3. Discussion

PCa is one of the commonest cancers in men worldwide, and diet is known as a causative factor in the induction and progression of PCa. Thus, it seems crucial to evaluate the possible effects of the presence of mycotoxins in one’s diet on prostate carcinogenesis and cancer progression. DON is discussed as one of the potential disruptors of the male reproductive system. It is postulated that the steroidogenesis-dysregulating effect of DON is associated with the modulation of steroidogenic enzymes [4] as well as germ cell degeneration, sperm retention, and abnormal morphology in male mice [24]. DON is also reported to have a toxic effect in the mice Leydig cell line MA-10 [33], affecting steroidogenesis and inducing oxidative stress [34].

The aim of this study was to investigate if a single exposure to DON in high doses could influence the basic molecular processes in PCa progression: cell viability, oxidative stress, apoptosis, and the cell cycle. Taking into consideration that androgens play a crucial role in PCa progression, different *in vitro* models of PCa were used in this study. We observed significant differences between androgen-dependent and androgen-independent cell lines in the response to DON. Firstly, it seems that androgen-independent cell lines (PC3 and DU-145) are more sensitive to DON: the same high doses triggered higher decreases in cell viability. Zhang et al. confirmed that ROS production was significantly increased after DON treatments in the porcine kidney cell line [35] and Li et al. showed that DON caused significantly increased (3.5 fold, compared to the control) ROS production in chicken embryo fibroblast DF-1 cells [1]. We observed that DON (2–5 µM) causes an increase in ROS production, a decrease in SOD2 antioxidant activity and expression as well as a significant increase in the expression of genes associated with the NFKB and HIF-1α signaling pathway. DON was previously reported to modulate HIF-1α expression in the process of oxidative stress in chickens [36] and constitutively modulates the expression of NFΚB in the inhibition of cancer cell growth [37]. In human gastric epithelial cells, the toxicity of DON was associated with the modulation of apoptosis regulated by FOXO3a [30]. Our results are consistent with these reports and present a significant increase in the expression of *RelA*, *HIF-1α,* and *FOXO3a* after exposure of PCa cells to DON. We also observed that DON induces apoptosis in the 22Rv1 and LNCaP cells, and it also modulates mitochondrial potential. In the investigated androgen-dependent PCa cells, DON caused a significant increase in the number of live and depolarized cells, whereas no decrease in dead depolarized cells was observed, as in the case of the androgen-independent DU-145 cell line. Previously, Bensassi et al. demonstrated that DON reduced the mitochondrial transmembrane potential in human colon carcinoma cells [35,38].

The androgen-independent PCa cell lines PC3 and DU-145 presented distinctive responses to DON treatment. Similar to what occurs in the 22Rv1 and LNCaP cells, DON caused an increase in ROS production and the number of dead cells with depolarized mitochondria as well as an increase in the number of apoptotic cells. These changes were associated with an increased expression of pro-apoptotic markers (cleaved Casp3 and p53). Interestingly, a distinct effect was observed in the PC3 cells. Although we observed an increase in ROS production, depolarization of the mitochondria and an increase in the expression of *RelA, HIF-1α,* and *FOXO3a*, apoptotic cells were not observed. This fact was also associated with a decrease in p53, cleaved Casp3 and PARP expression. We did not observe an increase in p21 expression (*CDKN1A*), whose expression was significantly increased for other cell lines. This fact is probably associated with the documented mutated form of p53 in PC3 cells which does not enable the activation and execution of apoptosis by p21. Bondy et al. suggests that p53 plays no role in DON-induced toxicity in mice [39]. We also observed that the lack of apoptosis in the PC3 cells did not decrease cell viability or ROS production. Moreover, for all tested cell lines we observed a significant increase in DNA damage. As reported before, ROS production and the oxidative damage of DNA and mitochondria are the main molecular effects of DON toxicity in cells [40]. Based on this, it is possible that p53 plays a crucial role in DON-induced apoptosis in cells, which might be a cause of oxidative stress, but this statement needs further studies to be confirmed.

We also observed that DON modulates the distribution of cells during the cell cycle. For all cell lines, we observed a decrease in the number of cells in the G0/G1 cell cycle phase with a simultaneous increase in the number of cells in the S and G2/M cell cycle phase. A similar increase in the number of cells in G2/M cell cycle phase was observed by Yuan et al. in HepG2 cells treated with DON at doses of 2 and 4 µg/mL [26]. The changes in the cell cycle caused by DON were also observed by Lei at al., though they observed the cell cycle arrest in the G0/G1 cell cycle phase in human chondrocyte, hepatic epithelial and tubular epithelial cells [41]. The changes in the distribution of the cell cycle after DON exposure were also visible in the expression of regulators of the progression of the G1 cell cycle phase: *CCND1* and *CDK4*. A similar increase in *CCND1* and *CDK4* expression caused by DON was observed by Mishra et al. in human keratinocytes [42].

The MAPK signal transduction pathway is highly conserved and plays a critical role in cell growth, differentiation, apoptosis, proliferation, and the response to environmental stress [43]. The results of this study showed that DON activates the MAPK signaling pathway, the phospho-ERK and the phospho-JNK in a concentration-dependent manner. DON is reported to modulate MAPKs in human immune cells [44] and in porcine epithelial cells [45,46]. It was previously observed that DON modulates the MAPK signaling pathway, and this effect is associated with the induction of apoptosis in piglet hippocampal nerve cells [47]. It is also possible that the observed effect of DON in PCa cells is associated with the MAPK signaling pathway.

## 4. Conclusions

Our findings indicate that a single exposure to high concentration of DON influences the viability of PCa cells by increasing of ROS production and DNA damage. The effect of DON might be associated with the androgen-sensitivity of cells, but needs further studies to be confirmed. DON is also able to induce apoptosis in PCa cells, but it seems that the presence of p53 in cells plays a crucial role in this process. Moreover, we observed that the effect of DON on PCa cells is associated with MAPK signaling, as well as the activation of FOXO3a and NFκB-HIF-1α-associated signaling pathways.

## 5. Materials and Methods

### 5.1. Cell Culture and DON Exposure

Human prostate adenocarcinoma cell lines LNCaP, PC3, and 22Rv1 were purchased from the European Collection of Authenticated Cell Cultures (ECACC) (Sigma Aldrich, Saint Louis, MO, USA) and DU-145 from the American Type Culture Collection (ATCC, Manassas, VA, USA). Cells were cultured in RPMI (LNCaP, PC3, 22Rv1) or Dulbecco’s modified Eagle’s medium (DMEM) (DU-145) supplemented with 10% heat-inactivated fetal bovine serum (FBS), 2 mM L-glutamine, 1 mM sodium pyruvate, 10 mM 4-(2-hydroxyethyl)-1-piperazineethanesulfonic acid buffer (HEPES) together with 50 U/mL penicillin and 50 μg/mL streptomycin (Thermo Fisher Scientific Inc, Waltham, MA, USA), in a humidified atmosphere of 5% CO_2_ at 37 °C. To avoid any possible involvement of serum components, the cells were treated without antibiotics and serum.

Stock solution of deoxynivalenol (DON) (Sigma-Aldrich, Saint Louis, MO, USA) was prepared in ethanol. The cells were treated with ethanol at the highest concentrations used in the study and no statistically significant decrease in viability was observed. The final concentrations of DON were achieved by adding culture medium. Cells were treated with DON for 24 h. Non-treated cells were used as controls.

### 5.2. Cell Viability

Cell viability was evaluated by the MTT test (3-(4,5-Dimethylthiazol-2-yl)-2,5-Diphenyltetrazolium Bromide) (MTT) (Merck Millipore, Burlington, MA, USA) according to the manufacturer’s instructions. 15 × 10^4^ cells were seeded on 96-well plates and incubated to reach 90% confluence. Next, cells were treated with experimental medium containing 0.0001–40 μM DON for 24 h. Four hours prior to the end of the incubation period, 5 mg/mL MTT solution was diluted to a final concentration of 0.5 mg/mL in each well. The formazan crystals formed by viable cells were dissolved in 10% SDS with 0.01 M HCl (100 μL), incubated overnight and measured using an ELX 808IU microplate reader (BioTek, Winooski, VT, USA) at 570 nm. The results are expressed as a percentage of non-treated cells. Each experiment was conducted in six replications.

### 5.3. Morphological Investigation

Human prostate adenocarcinoma cell lines were seeded on 12-well plates and treated with DON (0.1–5 µM) for 24 h. After treatment, the cells were photographed with an Olympus CKX41 microscope (Olympus, Tokyo, Japan) with an Olympus DP20 digital camera.

### 5.4. Cell Cycle Analysis

Propidium iodide (PI) staining in the presence of RNAse was used to evaluate the percentage of cells in the G0/G1, S and G2 phase of cell cycle with the Muse^®^ Cell Cycle Assay Kit (Merck Millipore, Burlington, MA, USA). Cells (3 × 10^5^/well) were seeded on 6-well plates and cultured to reach 90% confluence. Next, treated with experimental media containing 2 μM, 3 μM and 5 μM of DON for 24 h and trypsinized. The Cell Cycle Assay was conducted according to manufacturer’s recommendations. Cells were analyzed on Muse™ Cell Analyzer (Merck Millipore, Burlington, MA, USA) and compared to control cells. The results were expressed as the percentage of cells in each cell cycle phase. The experiment was conducted in triplicate.

### 5.5. Analysis of Changes in Mitochondrial Transmembrane Potential

Muse™ MitoPotential Assay (Merck Millipore, Burlington, MA, USA) which evaluates the polarization of mitochondria was used to measure the changes in mitochondrial membrane potential (ΔΨm) caused by DON. Simultaneous use of 7-AAD allows evaluation of cell membrane integrity. Cells (approximately 8 × 10^5^/well) were seeded on 6-well plates and left to reach 90% confluence. Then, the cells were treated with experimental medium containing 2 μM, 3 μM, and 5 μM of DON for 24 h. The assay was performed as recommended by the manufacturer. The probes were standardized against control probes. The experiment was repeated three times.

### 5.6. Annexin V and Dead Cell Double Staining Assay

PCa cells apoptosis was determined using the Muse™ Annexin V and Dead Cell Kit (Merck Millipore, Burlington, MA, USA). The cells (3 × 10^5^/well) were seeded on 6-well plates and left to reach 90% confluence. After exposure to DON (2 μM, 3 μM, and 5 μM), the cells were detached and suspended in 100 μL of culture medium. The assay was performed according to the manufacturer’s instructions. The analysis was performed in three independent experiments.

### 5.7. Multi-Color DNA Damage

The effect of DON on DNA damage was determined with the Muse™ Multi-Color DNA Damage Kit (Merck Millipore, Burlington, MA, USA). Cells (4 × 10^5^/well) were seeded on 12-well plates and treated with experimental medium containing 2 μM, 3 μM, and 5 μM of DON for 24 h. The assay was performed according to the manufacturer’s instructions. The analysis was performed in three independent experiments.

### 5.8. Morphology of Mitochondria and Nuclei

To evaluate the morphology of nuclei 4′,6-diamidino-2-phenylindole (DAPI, Life Technologies, Carlsbad, CA, USA) fluorescent dye was used. Human prostate adenocarcinoma cell lines were seeded on 96-well plates were treated with DON (2–5 µM) for 24 h. For DNA staining, the cells were fixed with 70% ethanol and stained with 150 nM DAPI in PBS. FLoid Cell Imaging Station (Thermo Fisher Scientific Inc, Waltham, MA, USA) was used to capture images.

### 5.9. Evaluation of Reactive Oxygen Species (ROS) Levels

Cells (5 × 10^5^/well) were seeded on six-well plates and cultured to reach 90% confluence. Then were treated with experimental medium containing 2 μM, 3 μM, and 5 μM of DON for 24 h. Oxidative Stress Assay (Merck Millipore, Burlington, MA, USA) was performed according to the manufacturer’s instructions. The probes were measured on a Muse™ Cell Analyzer (Merck Millipore, Burlington, MA, USA) and standardized against control probes. The analysis was performed in three independent experiments.

### 5.10. Measurement of Cu/Zn Superoxide Dismutase (SOD1) and Glutathione Peroxidase (GPx) Activity

Cells (4 × 10^5^) were seeded on 100 mm Petri dishes and left to reach 90% confluence. Cells were treated with DON for 24 h. Superoxide Dismutase (SOD) Colorimetric Activity ELISA Kit (Thermo Fisher Scientific Inc, Waltham, MA, USA) was conducted according to the manufacturer’s protocol. The absorbance was measured using an EL808IU BioTek microplate reader (BioTek, Winooski, VT, USA) at 450 nm. The results for the assay were expressed as U/mL, based on the standard curve. The probes were measured in two repeats.

Cells (1 × 10^6^) were seeded on 60 mm Petri dishes and left to reach 90% confluence and then treated with experimental medium containing 2 μM, 3 μM, and 5 μM of DON for 24 h. (GPx) content was determined by Glutathione Peroxidase Assay Kit (Abcam, Cambridge, UK) according to the manufacturer’s instructions. Absorbance was measured at 340 nm. The probes were measured in duplicate.

### 5.11. Expression of Apoptotic Proteins

PathScan^®^ Apoptosis Multi-Target Sandwich ELISA Kit #7105 (Cell Signalling Technology, Leiden, WZ, The Netherlands) was used to analyze the expression of key signaling proteins associated with apoptosis (p53, p-p53, cleaved Casp3, PARP). The assay was performed according to the manufacturer’s recommendations. The concentration of protein used in experiment was 1 mg/mL for 22Rv1, LNCaP, DU-145 cells and 0.9 mg/mL for PC3.

### 5.12. Real Time quantitative Polymerase Chain Reaction (RT-qPCR)

For RNA isolation, (1 × 10^6^) cells were cultured on 60 mm Petri dishes and treated with experimental medium containing 2 μM, 3 μM, and 5 μM of DON for 24 h. Cells were then suspended in TRIzol Reagent (Thermo Fisher Scientific Inc, Waltham, MA, USA) and RNA was isolated according to the manufacturer’s protocol. The concentration of isolated RNA diluted in 50 μL of sterile deionized water was measured with a BioDrop DUO spectrophotometer (BioDrop, Cambridge, UK). 5μg of total RNA was used to synthesize cDNA using ImProm RT-IITM reverse transcriptase (Promega, Madison, WI, USA) according to the manufacturer’s instructions. RT-qPCR was conducted with LightCycler 96 (Roche, Basel, Switzerland) with 2μL of cDNA. Primers were designed and verified using Primer-BLAST (National Institutes of Health) (Table 5). The Human Reference RNA (Stratagene, San Diego, CA, USA) was used as a calibrator. The relative expressions of cyclin B1 (*CCNB1*), cyclin-dependent kinase 1 (*CDC2*), cyclin dependent kinase 4 (*CDK4*), cyclin dependent kinase inhibitor 1A (*CDKN1A*), cyclin D1 (*CCND1*), caspase 3 (*Casp3*), BCL2 associated X, apoptosis regulator (*Bax*), superoxide dismutase 1 (*SOD1*), superoxide dismutase 2 (*SOD2*), hypoxia inducible factor 1α (*HIF-1α*), forkhead box O3 (*FOXO3*), transcription factor p65 (*RelA*). Ribosomal protein S17 (*RPS17*), ribosomal protein P0 (*RPLP0*), and histone H3.3A (*H3F3A*) were used as a reference genes. The melting curve analysis was performed to confirm the specificity of the product for each primer set. The data was analyzed with the ΔΔCt method. Each reaction was conducted in a duplicate of three repeats of the experiment and expressed as a relative expression.

### 5.13. Western Blot Analysis

For the protein isolation, 1 × 10^6^ cells were cultured on 100mm Petri dishes and treated with experimental medium containing of DON for 24 h. The protein isolation and Western blots were conducted as previously described [48]. 30 µg (60 µg for phospho-p44/42 (Thr202/Tyr204), phospho-p38 (Thr180/Tyr182), phospho-SAPK/JNK (Thr183/Tyr185) antibody) of protein samples was used for electrophoresis. SOD1 (#4266), SOD2 (#13141), p65 (#8242), p44/42 (#4695), SAPK/JNK (#9252), p38 (#8690), phospho-p44/42 (Thr202/Tyr204) (#4370), phospho-p38 (Thr180/Tyr182) (#4511), phospho-SAPK/JNK (Thr183/Tyr185) (#4668) (Cell Signaling Technology, Leiden, WZ, The Netherlands) antibodies were used according to manufacturer’s recommendations. The results were normalized to glyceraldehyde 3-phosphate dehydrogenase (GAPDH) (sc-59540) (Santa Cruz Biotechnology Inc, Dallas, TX, USA) as a reference protein.

### 5.14. Statistical Analysis

Results were expressed as mean ± SE. The one-way ANOVA test was used to calculate statistically significant differences. *p* < 0.05 was considered as statistically significant. GraphPad Prism software (GraphPad Software, La Jolla, CA, USA) was used to carry out all statistical analyses.

## Figures and Tables

**Figure 1 toxins-11-00265-f001:**
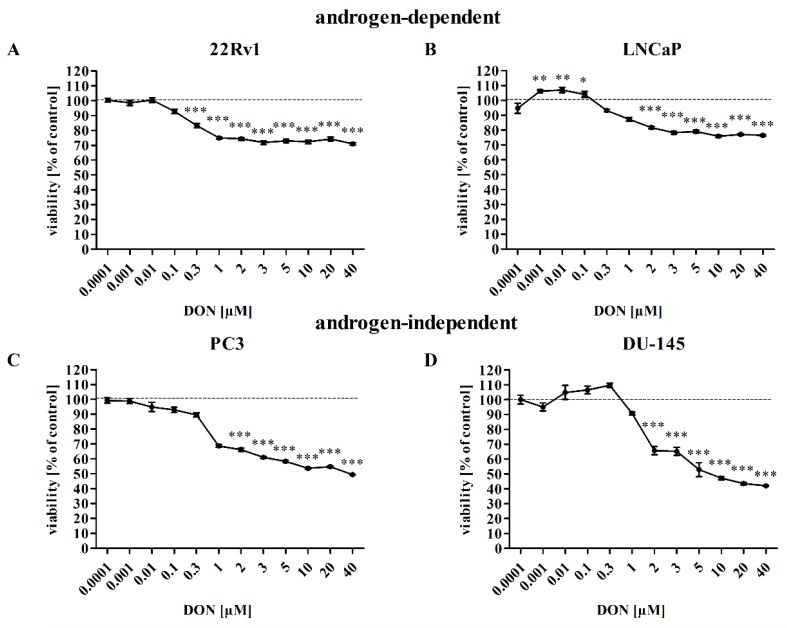
DON decreases the viability of prostate cancer cells. The results were obtained by MTT assay and expressed as a percentage of control cells for 22Rv1 (**A**), LNCaP (**B**), PC3 (**C**), and DU-145 (**D**) cells. Changes in cell morphology after 24 h of DON treatment (**E**). *p* < 0.05 was considered statistically significant, **** p* < 0.001 as compared to the control. DON—deoxynivalenol, Cnt—control.

**Figure 2 toxins-11-00265-f002:**
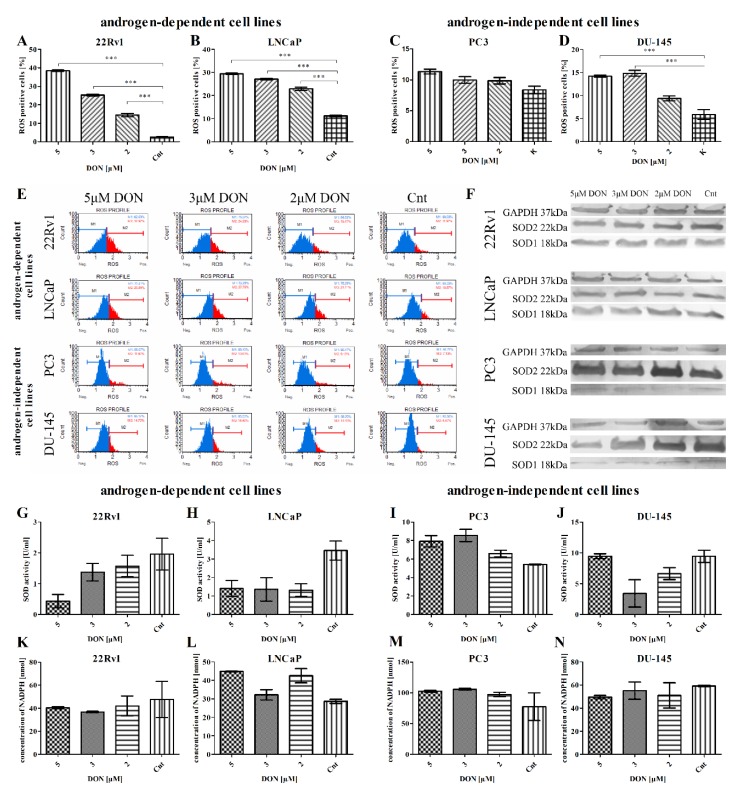
DON induces oxidative stress in prostate cancer cells. The changes in ROS production in 22Rv1 (**A**), LNCaP (**B**), PC3 (**C**), and DU-145 (**D**) cells after DON treatment for 24 h were expressed as the percentage of gated cells; non-treated cells were used as a control; the values are expressed as the mean ±SE; *p* < 0.05 was considered statistically significant, **** p* < 0.001 as compared to the control. The representative histogram of cells distribution according to ROS production after DON treatment (**E**). The representative results of Western blot analysis (**F**). The activity of SOD and GPx enzymes in 22Rv1 (**G**,**K**), LNCaP (**H**,**L**), PC3 (**I**,**M**) and DU-145 (**J**,**N**) cells was estimated by ELISA. SOD1—superoxide dismutase 1, SOD2—superoxide dismutase 2, GAPDH—anti-glyceraldehyde 3-phosphate dehydrogenase, GPx—glutathione peroxidase, DON—deoxynivalenol, Cnt—control.

**Figure 3 toxins-11-00265-f003:**
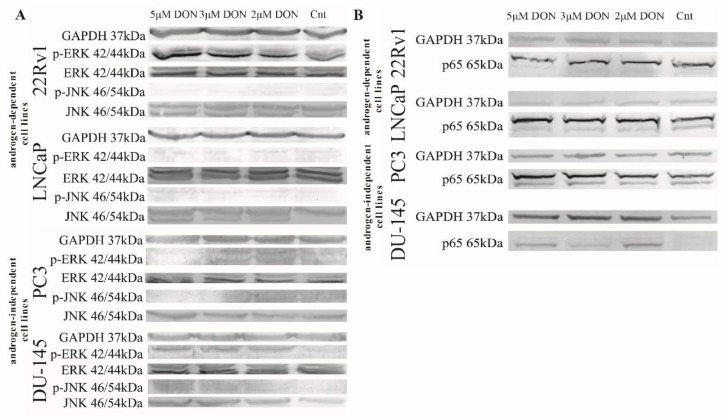
DON modulates MAPK and p65 expression in prostate cancer cell lines. The expression of mitogen-activated protein kinase (MAPK) after DON treatment (**A**). The expression of p65 after DON treatment (**B**). The results were obtained by Western blot. GAPDH was used as a reference. DON—deoxynivalenol, Cnt—control.

**Figure 4 toxins-11-00265-f004:**
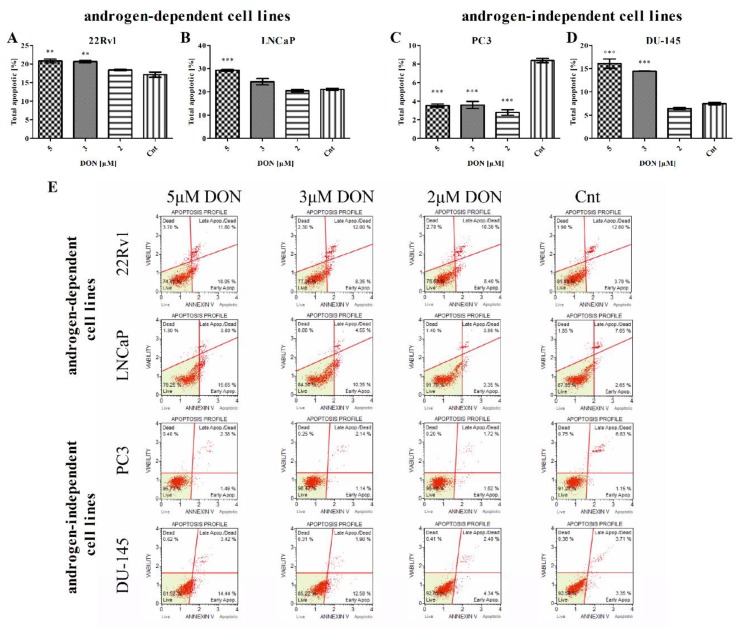
DON induces apoptosis in prostate cancer cells. The changes in the number of apoptotic (early and late apoptotic) in 22Rv1 (**A**), LNCaP (**B**), PC3 (**C**), and DU-145 (**D**) cells after DON treatment were measured and expressed as the percentage of gated cells; non-treated cells were used as controls; the values are expressed as the mean ± SE; *p* < 0.05 was considered statistically significant, **** p* < 0.001, ** *p* < 0.001, * *p* < 0.05 as compared to the control. The representative results of the Apoptosis Profile after DON treatment (**E**). DON—deoxynivalenol, Cnt—control.

**Figure 5 toxins-11-00265-f005:**
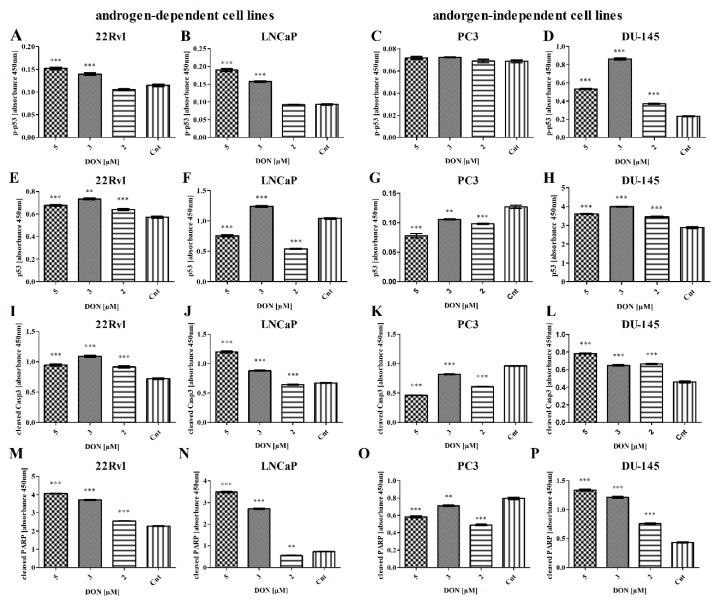
DON modulates the expression of p53, p-p63, cleaved Casp3, and PARP in 22Rv1 (**A**,**E**,**I**,**M**), LNCaP (**B**,**F**,**J**,**N**), PC3 (**C**,**G**,**K**,**O**), and DU-145 (**D**,**H**,**L**,**P**) cells. The results of ELISA are presented as absorbance values. *p * <  0.05 was considered statistically significant, **** p  *<  0.001 *** p * <  0.01, as compared to the control cells. DON—deoxynivalenol, Cnt—control.

**Figure 6 toxins-11-00265-f006:**
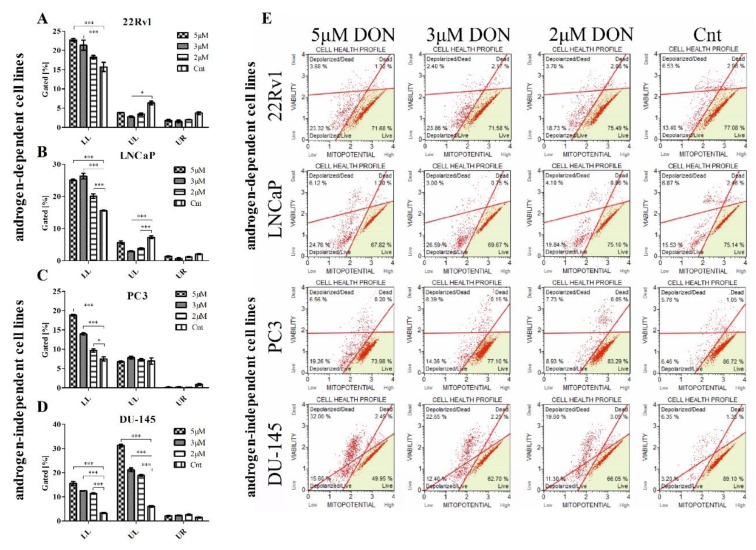
The influence of DON on the mitochondrial potential of 22Rv1 (**A**), LNCaP (**B**), PC3 (**C**), and DU-145 (**D**) cells. The changes in the number of live cells with depolarized mitochondria (LL), dead cells with depolarized mitochondria (UL) and dead cells with mitochondria that had not been depolarized (UR) after DON treatment were estimated by the Muse™ MitoPotential Assay; non-treated cells were used as a control; the values are expressed as the mean ± SE; *p* < 0.05 was considered statistically significant, **** p* < 0.001, * *p* < 0.05 as compared to the control. A representative histogram of the cell health profile after DON treatment (**E**). DON—deoxynivalenol, Cnt—control.

**Figure 7 toxins-11-00265-f007:**
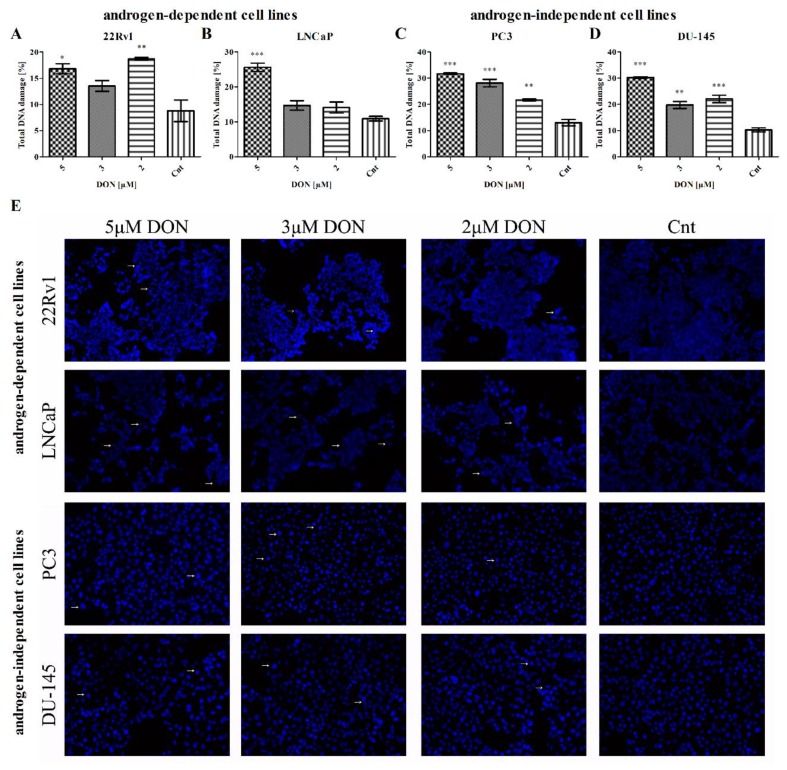
The influence of DON on DNA damage. The changes in the percentage of cells with damaged DNA after DON treatment in 22Rv1 (**A**), LNCaP (**B**), PC3 (**C**), and DU-145 (**D**) were expressed as the percentage of gated cells; non-treated cells were used as a control; the values are expressed as the mean ± SE; *p* < 0.05 was considered statistically significant, **** p* < 0.001, *** p* < 0.001, ** p* < 0.05 as compared to the control. DAPI stained nuclei (**E**) after DON treatment, and non-proper nuclei morphology was marked with white arrows; non-treated cells were used as the control. DON—deoxynivalenol, Cnt—control.

**Figure 8 toxins-11-00265-f008:**
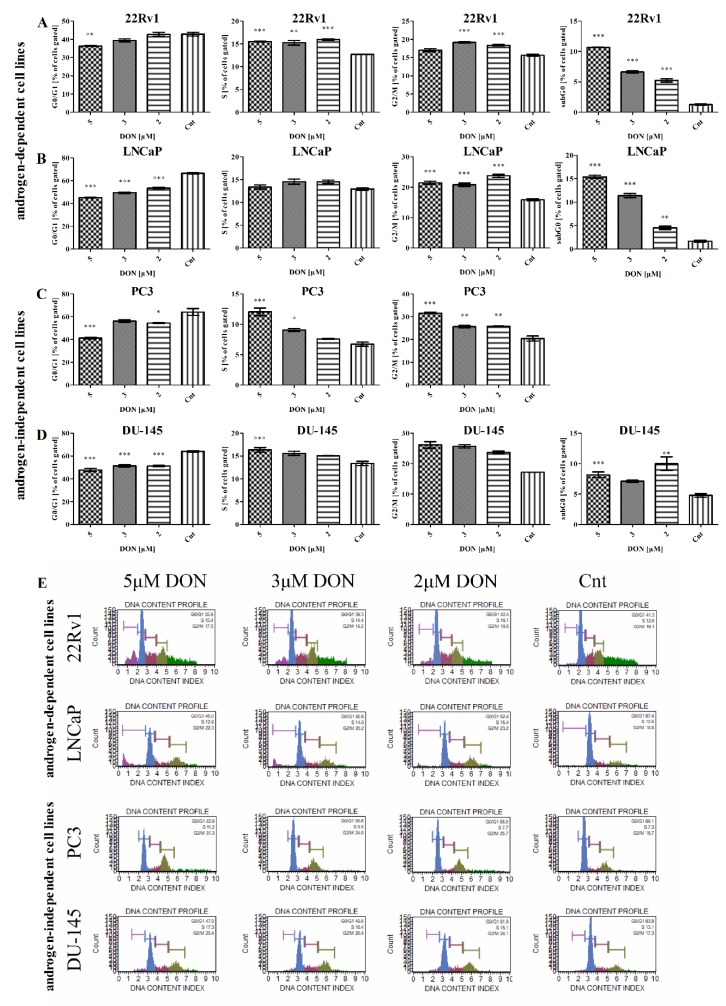
The influence of DON on cell cycle progression. The number of 22Rv1 (**A**), LNCaP (**B**), PC3 (**C**), and DU-145 (**D**) cells in the G0/G1, S and G2/M cell cycle phases was expressed as percentage of gated cells. The results are expressed as the mean ± SE. The representative results obtained by flow cytometry with the Cell Cycle Analysis kit (**E**). *p* < 0.05 was considered statistically significant, **** p* < 0.001, ** *p* < 0.001, * *p* < 0.05 as compared to the control. DON—deoxynivalenol, Cnt—control.

**Table 1 toxins-11-00265-t001:** The relative expression of SOD1 and SOD2 after DON treatment. The results are presented as a mean. *p* < 0.05 was considered statistically significant, **** p* < 0.001, *** p* < 0.001, * *p* < 0.05, as compared to the control. SOD1—superoxide dismutase 1, SOD2—superoxide dismutase 2, DON—deoxynivalenol, Cnt—control.

Androgen Dependency	Cell Line	Dose	*SOD1*	*SOD2*
Androgen-dependent cell lines	22Rv1	5 µM DON	0.8216 ***	0.4710
		3 µM DON	0.9158 ***	0.5009
		2 µM DON	0.9122 ***	0.5504
		Cnt	0.6205	0.5013
	LNCaP	5 µM DON	1.038 ***	1.310 ***
		3 µM DON	1.009 ***	0.7880 ***
		2 µM DON	0.9110 **	0.5913 **
		Cnt	0.5760	0.1688
Androgen-independent cell lines	PC3	5 µM DON	1.104 ***	4.268 *
		3 µM DON	1.208 ***	3.135
		2 µM DON	1.241 ***	3.310
		Cnt	1.568	2.685
	DU-145	5 µM DON	0.8172 ***	1.030 ***
		3 µM DON	0.9245 ***	1.087 ***
		2 µM DON	1.061 *	1.314 ***
		Cnt	1.216	2.214

**Table 2 toxins-11-00265-t002:** The relative expression of genes associated with oxidative stress. The results are expressed as a mean. *p* < 0.05 was considered statistically significant, **** p* < 0.001, ** *p* < 0.001 as compared to the control. RelA—transcription factor p65, FOXO3—forkhead box O3, HIF1α—hypoxia inducible factor 1α, DON—deoxynivalenol, Cnt—control.

Androgen Dependency	Cell Line	Dose	*RelA*	*FOXO3*	*HIF1α*
Androgen-dependent cell lines	22Rv1	5 µM DON	4.940 ***	1.760 ***	1.152 ***
		3 µM DON	5.045 ***	1.300 ***	1.021 ***
		2 µM DON	5.015 ***	0.9127 ***	0.8682 ***
		Cnt	0.8955	0.4464	0.2178
	LNCaP	5 µM DON	9.409 ***	1.557 ***	1.557 ***
		3 µM DON	7.976 ***	1.075 ***	1.073 ***
		2 µM DON	7.424 ***	1.010 ***	1.012 ***
		Cnt	0.7651	0.2067	0.2066
Androgen-independent cell lines	PC3	5 µM DON	9.593 ***	5.230 ***	5.228 ***
		3 µM DON	8.874 ***	5.765 ***	5.765 ***
		2 µM DON	7.222 ***	4.850 ***	4.848 ***
		Cnt	3.803	3.245	3.246
	DU-145	5 µM DON	6.836 ***	1.499 ***	1.784 ***
		3 µM DON	5.495 ***	1.163 ***	1.867 ***
		2 µM DON	6.307 ***	1.125 ***	2.214 ***
		Cnt	2.17	0.4242	1.275

**Table 3 toxins-11-00265-t003:** The relative expression of genes associated with apoptosis. The results are expressed as a mean. *p* < 0.05 was considered statistically significant, **** p* < 0.001, *** p* < 0.001, * *p* < 0.05 as compared to the control. Casp3—caspase 3, Bax—BCL2 associated X apoptosis regulator, DON—deoxynivalenol, Cnt—control.

Androgen Dependency	Cell Line	Dose	*Casp3*	*Bax*
Androgen-dependent cell lines	22Rv1	5 µM DON	2.080 ***	1.372 **
		3 µM DON	1.713 ***	1.392 **
		2 µM DON	1.416 ***	1.353 *
		Cnt	0.3414	0.8033
	LNCaP	5 µM DON	3.850 ***	1.407 **
		3 µM DON	2.927 ***	1.242 *
		2 µM DON	2.430 ***	1.052
		Cnt	0.4825	0.7100
Androgen-independent cell lines	PC3	5 µM DON	1.756 **	0.7383 *
		3 µM DON	1.800 ***	0.8233
		2 µM DON	1.687 *	0.9817
		Cnt	1.367	1.033
	DU-145	5 µM DON	1.048	1.128 ***
		3 µM DON	0.8198	0.8817
		2 µM DON	1.036	1.170 ***
		Cnt	0.9308	0.6750

**Table 4 toxins-11-00265-t004:** The relative expression of genes associated with the cell cycle. The results are presented as a mean. *p* < 0.05 was considered statistically significant, **** p* < 0.001, ** *p* < 0.001, * *p* < 0.05 as compared to the control. *CCND1*—cyclin D1, *CDK4*—cyclin dependent kinase 4, *CDC2*—cyclin-dependent kinase 1, *CCNB1*—cyclin B1, *CDKN1A*—cyclin dependent kinase inhibitor 1A, DON—deoxynivalenol, Cnt—control.

Androgen Dependency	Cell Line	Dose	*CCND1*	*CDK4*	*CDC2*	*CCNB1*	*CDKN1A*
Androgen-dependent cell lines	22Rv1	5 µM DON	6.570	1.727 ***	1.148	0.5915	1.928 ***
		3 µM DON	8.919 ***	1.489 **	0.8959	0.5901	2.017 ***
		2 µM DON	11.33 ***	1.441 **	0.8747	0.6257	1.914 ***
		Cnt	4.163	1.024	0.8164	0.3827	0.6593
	LNCaP	5 µM DON	1.669 ***	1.001 ***	1.303 **	0.9518 ***	30.80 ***
		3 µM DON	1.338 ***	0.9578 ***	0.8296	0.5194	28.24 ***
		2 µM DON	1.363 ***	0.9136 ***	1.208 *	0.9653 ***	22.53 ***
		Cnt	0.6477	0.5710	0.6072	0.2413	2.946
Androgen-independent cell lines	PC3	5 µM DON	8.594 ***	1.639 ***	2.603 ***	1.007	1.071
		3 µM DON	8.000 ***	1.909 ***	2.948 **	0.8900 ***	0.9634
		2 µM DON	7.236 **	1.781 ***	2.932 **	0.8485 ***	0.9790
		Cnt	4.831	1.557	3.811	1.164	1.250
	DU-145	5 µM DON	2.417 ***	1.036 **	1.445 ***	0.8115 ***	4.855 ***
		3 µM DON	1.748 ***	0.9628 **	1.707 ***	1.178	3.503 ***
		2 µM DON	2.102 ***	1.003 **	1.928 *	1.786 ***	3.131 ***
		Cnt	0.7869	0.7160	1.121	1.235	0.9874 ***

**Table 5 toxins-11-00265-t005:** Primers used in RT-qPCR. *CCNB1*—cyclin B1, *CDC2*—cyclin-dependent kinase 1, *CDK4*—cyclin dependent kinase 4, *CDKN1A*—cyclin dependent kinase inhibitor 1A, *CCND1*—cyclin D1, *Casp3*—caspase 3, *Bax*—BCL2 associated X, apoptosis regulator, *SOD1*—superoxide dismutase 1, *SOD2*—superoxide dismutase 2, *HIF-1α*—hypoxia inducible factor 1α, *FOXO3*—forkhead box O3, *RelA*—transcription factor p65, *RPS17*—ribosomal protein S17, *RPLP0*—ribosomal protein P0, *H3F3A*—histone H3.3A.

Gene	Sequence (5′-3′)	Product Size (bp)
***CCNB1***	For ACCTATGCTGGTGCCAGTG Rev GGCTTGGAGAGGCAGTA	128
***CDC2***	For TTTTCAGAGCTTTGGGCACT Rev AGGCTTCCTGGTTTCCATTT	100
***CDK4***	For TTACTGAGGCGACTGGAGG Rev GTCCTTAGGTCCTGGTCTACAT	131
***CDKN1A***	For GACAGATTTCTACCACTCCAA Rev CTGAGACTAAGGCAGAAGAGT	134
***CCND1***	For TGTCCTACTACCGCCTCACACGCTTCCTCTCCAG Rev TCCTCTTCCTCCTCCTCGGCGGCCTTG	160
***Casp3***	For GGAATATCCCTGGACAACAGTT Rev TTGCTGCATCGACATCTGT	130
***Bax***	For AGAGGTCTTTTTCCGAGTGGCAGC Rev TTCTGATCAGTTCCGGCACCTTG	137
***SOD1***	For GCGTGGCCTAGCGAGTTAT Rev ACACCTTCACTGGTCCATTACT	114
***SOD2***	For GGGTTGGCTTGGTTTCAATAAG Rev CTGAAGGTAGTAAGCGTGCTC	136
***HIF-1α***	For TTACTCATCCATGTGACCATGA Rev AGTTCTTCCTCGGCTAGTTAG	140
***FOXO3***	For CAAGGATAAGGGCGACAG Rev GGTTGATGATCCACCAAGA	131
***RelA***	For GCACAGATACCACCAAGACC Rev TCAGCCTCATAGAAGCCATC	157
***RPS17***	For AAGCGCGTGTGCGAGGAGATCG Rev TCGCTTCATCAGATGCGTGACATAACCTG	87

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
