# Peer review of "Deoxynivalenol Modulates the Viability, ROS Production and Apoptosis in Prostate Cancer Cells"

_toxins, 2019, doi:10.3390/toxins11050265_

Round 1

Reviewer 1 Report

I highly appreciated the effort of the authors for significantly improving the manuscript.

I agree with the answer at technical issues.

Author Response

Dear Reviewer,

We are thankful for your support during revision and resubmission of the manuscript.

Reviewer 2 Report

The present manuscript is relatively interesting in terms of hormone-related differential responses in cancer cells. There are several critical major points which should be addressed for mechanism and rationale.

1.  What makes the differential responses to DON b/w Androgen-dependent cells and independent cells. Author can provide evidences among signaling mediators (mapk. hif, foxo, p53).

2. Author presented different types of readouts for biological markers on apoptosis, ROS, SOD, cell cycle arrest). However, only these events are not new things since there have been a lot of published reports. For more scientific priority, these events (apoptosis, radical production, cell cycle arrest)  need to to be mechanistically addressed using blocker of the provided molecular signaling events in different hormone-related cell lines. 

3. Quality of western blots are so bad to tell and quantify the activation in spite of they are representative. Strongly recommended for better blots by performing again.

4.  Quality of DAPI imaging are so bad to tell and quantify the DNA FRAGMENTATION in spite of they are representative. Strongly recommended for better resolution images.  

Author Response

Dear Reviewer,

Please find below the answers to your comments.

1.  What makes the differential responses to DON b/w Androgen-dependent cells and independent cells. Author can provide evidences among signaling mediators (mapk. hif, foxo, p53). The aim of this study was to evaluate the effect of DON on prostate cancer cells. We used different in vitro models of prostate cancer to evaluate the response of cells thoroughly. We observed that the sensitivity of cells is different, especially the response of PC3 cells, which does not possess a proper p53 form. The manuscript has been revised as requested by the Reviewers in terms of androgen sensitivity of cells. In our opinion the lack of p53 is the reason of different response (the lack of apoptosis), what was clearly explained in the manuscript. In the manuscript we also evaluated the expression of HIF1α, FOXO3a, RelA as well as ERK and JNK as a part of MAPK signaling pathway.

2. Author presented different types of readouts for biological markers on apoptosis, ROS, SOD, cell cycle arrest). However, only these events are not new things since there have been a lot of published reports. For more scientific priority, these events (apoptosis, radical production, cell cycle arrest)  need to  be mechanistically addressed using blocker of the provided molecular signaling events in different hormone-related cell lines. As far as we are concerned, there is no studies evaluating the effect of DON on prostate cancer cells, especially mimicking a single exposure to DON.  We believe that different methods already used in the manuscript to evaluate the oxidative stress, apoptosis and cell cycle control: cytometry as well as expression of genes and protein, supports  enough the results and are reliable. We are also sure that this manuscript will be a begging of research studies evaluating in detail the possible effect of DON on prostate cancer cells and carcinogenesis in in vivo studies.

3. Quality of western blots are so bad to tell and quantify the activation in spite of they are representative. Strongly recommended for better blots by performing again.  The representative results of Western blots are provided in higher resolution that is required (600dpi instead of 300dpi). The Western blots were conducted according to the validated procedure, the specificity of antibodies was also evaluated. The bands are clearly visible (we encourage to magnitude the figure). We do not practice to change the intensity of bands in any software, to provide a reliable results, thus bands are not black on the white background.

4.  Quality of DAPI imaging are so bad to tell and quantify the DNA FRAGMENTATION in spite of they are representative. Strongly recommended for better resolution images.  The fragmentation of DNA in DAPI staining is clearly visible by the shape of the nuclei of cells: in case of normal cells the nucleus is round shaped with constant staining. In case of fragmentation of DNA the nucleus of cells is fragmentated and stained patchily. It is visible without counting. The examples of fragmentated nuclei were marked in the figures with white arrows to make it easier for the reader to find out. All the figures were provided in higher resolution (600dpi) that is required (300dpi). The problem with the visibility of figures might be thus associated with the version of software.

We would like to also add that after the last submission, the manuscript  was edited by a native speaker from Language Division.

Reviewer 3 Report

The authors have made all the revisions suggested

Author Response

Dear Reviewer,

We are thankful for your support during revision and resubmission of the manuscript.

This manuscript is a resubmission of an earlier submission. The following is a list of the peer review reports and author responses from that submission.

Round 1

Reviewer 1 Report

Dear authors,

The clarity of the manuscript has significantly improved. Nevertheless, clarity needs some more readjustments. See below some suggestions:

- in tables and figures divide the investigated cell lines according to their androgen dependence (apparently this is a key feature of the study and this should be made clear when readers follow your results);

- as you advance in your study, please do make references to results previously described in the manuscript. For instance, correlate your more sophisticated parameters with the results of the initial MTT assay, apoptosis and ROS production (these initial measures are more intuitive);

- explain results based on particular characteristics of cells (androgen dependence or p53 particularities);

- explain in more detail the significance of each investigated parameter. For instance you start talking about MAPK and inflammation and give results about FOXO3. 

- please do not repeat results in the "Discussion" chapter. Because you have very many results which are somehow difficult to follow, try in this chapter to organize and explain your findings in a more comprehensive way without repeating results.

English was also improved but a thorough revision by a native English speaker would be necessary.

I am so sorry about the last version of the manuscript with my comments that I have sent you. My Acrobat reader does not work well and I found that maybe you received only the highlights in the manuscript and not the detailed comments aimed at helping you to improve the paper.

I send you a new version of comments and I really hope that they will be useful to you. The study is really well conducted but needs more clarity.

I want to highlight some technical issues:

- serum was not present in your cellular samples during experiment. This may interfere with the effect of DON. Please explain why you designed your experiment in this way.

- genes encoding ribosomal proteins were used as reference genes. Are you sure this is OK in the context of DON?

Author Response

Dear Reviewer,

We are really thankful for your support during review process.

I would like to submit the revised manuscript „Deoxynivalenol modulates the viability, ROS production and apoptosis in prostate cancer cells” for possible publication in Toxins. All the changes made in the manuscript has been made in the “Track changes” option in word with the exception of Figures and Tables (exchanged for corrected). In Figure 1 the one of diagrams has been doubled accidentally, but the result section was correct, thus it was changed for the correct one. According to the suggestion of The Associated Editor the manuscript has been corrected to avoid the self-plagiarism, nevertheless the names of the cell lines, media, kits, a commonly used phrases like “play a crucial role” or “increase in the number of apoptotic cells” etc. has not been changed due to the fact that they are just a long term names or phrases and cannot be rephrased, unfortunately. In other places, the sentences were rephrased.  The manuscript has been revised by a native speaker. Please, find the answers to Reviewers comments below, in italics.

- in tables and figures divide the investigated cell lines according to their androgen dependence (apparently this is a key feature of the study and this should be made clear when readers follow your results); In all figures and tables the androgen-dependent and androgen-independent cell lines were marked as suggested.

- as you advance in your study, please do make references to results previously described in the manuscript. For instance, correlate your more sophisticated parameters with the results of the initial MTT assay, apoptosis and ROS production (these initial measures are more intuitive); Where it was marked, the references to previous results were added.

- explain results based on particular characteristics of cells (androgen dependence or p53 particularities); We hope that the results were corrected adequately.

- explain in more detail the significance of each investigated parameter. For instance you start talking about MAPK and inflammation and give results about FOXO3. The explanation for the evaluation of RelA, HIF1a and FOXO3a expression has been added to the manuscript.

- please do not repeat results in the "Discussion" chapter. Because you have very many results which are somehow difficult to follow, try in this chapter to organize and explain your findings in a more comprehensive way without repeating results. The discussion has been corrected.

English was also improved but a thorough revision by a native English speaker would be necessary. The manuscript has been revised by a native speaker.

I want to highlight some technical issues:

- serum was not present in your cellular samples during experiment. This may interfere with the effect of DON. Please explain why you designed your experiment in this way. The serum might itself trigger changes in evaluated parameters. It is a gold standard in our lab in mycotoxin research to do not use serum in mycotoxin research. Thus, similar control without serum is always used to compare the results.

- genes encoding ribosomal proteins were used as reference genes. Are you sure this is OK in the context of DON? Yes, we are. We tested the expression of these genes in non- treated cells and treated with DON and observed no changes in the Cq value.

Page 10, correction number 1:I This rocket shape does not look good. It is possible, but the fractions of cells were clearly divided in that way. The manufacturer recommends to adjust the setting to the fractions of cells.

Page 16, correction number 4: You do not know about all the parameters you investigated. It is true, nevertheless the control of the experiments it usually based on the viability test. If there is no impact of the solvent, non-treated cells were used. Similarly, we did in this experiments.

Page 16, correction number 12:  Please specify the software.  The software is provided with Muse Cell Counter by Merck Millipore and is a part of it, thus we believe it is not necessary to explain it in methods section

Page 17, correction number 3: Maybe cultivation and treatment of cells in the experiment might be mentioned only once in a distinct subchapter. We believed that it is easier to find information in detail in divided sections instead of sub-sections.

It was also suggested to exchange the abbreviation for control in Figures, but this abbreviation is consistent in our publications, thus it was not changed.

Reviewer 2 Report

The manuscript extensively addressed diverse events in prostate cancer cells.

Despite the extensive observations, author provided only speculative mechanistic evidences  for the key outcomes (apoptosis, oxidative stress, cell cycle arrest,  ..).  

"The observed effect of DON in PCa cells might be associated with mitogen-activated protein kinase (MAPK) and nuclear factor kappa-light-chain-enhancer of activated B cells (NFΚB)- hypoxia inducible factor 1α (HIF-1α) signaling pathways. Our results also showed that p53 might be a crucial factor in DON-associated apoptosis in PCa cells" 

To overcome these demerits, all the key outcomes (apoptosis, oxidative stress, cell cycle arrest) need to be compared simply when blocking of MAPK signaling, NF-kB, and p53 pathway. 

Based on the confirmed evidences, different responses of different prostate cancer cells can be explained.

Author Response

Dear Reviewer,

I would like to submit the revised manuscript „Deoxynivalenol modulates the viability, ROS production and apoptosis in prostate cancer cells” for possible publication in Toxins. All the changes made in the manuscript has been made in the “Track changes” option in word with the exception of Figures and Tables (exchanged for corrected). In Figure 1 the one of diagrams has been doubled accidentally, but the result section was correct, thus it was changed for the correct one. According to the suggestion of The Associated Editor the manuscript has been corrected to avoid the self-plagiarism, nevertheless the names of the cell lines, media, kits, a commonly used phrases like “play a crucial role” or “increase in the number of apoptotic cells” etc. has not been changed due to the fact that they are just a long term names or phrases and cannot be rephrased, unfortunately. In other places, the sentences were rephrased.  The manuscript has been revised by a native speaker. Please, find the answers to Reviewer comments below, in italics.

“The manuscript extensively addressed diverse events in prostate cancer cells. Despite the extensive observations, author provided only speculative mechanistic evidences  for the key outcomes (apoptosis, oxidative stress, cell cycle arrest,  ..).  "The observed effect of DON in PCa cells might be associated with mitogen-activated protein kinase (MAPK) and nuclear factor kappa-light-chain-enhancer of activated B cells (NFΚB)- hypoxia inducible factor 1α (HIF-1α) signaling pathways. Our results also showed that p53 might be a crucial factor in DON-associated apoptosis in PCa cells" To overcome these demerits, all the key outcomes (apoptosis, oxidative stress, cell cycle arrest) need to be compared simply when blocking of MAPK signaling, NF-kB, and p53 pathway.”

The aim of this study is to evaluate the effect of DON on the viability, oxidative stress and apoptosis in different prostate cancer cell lines with different androgen sensitivity. We also suggested some molecular mechanism, which we believed might be associated with observed results, but this statement, as you said needs further studies to be confirmed. We believe that this study will be the first step to evaluate in detail (both in in vitro as well as in vivo models) the detailed effect of DON on human carcinogenesis.

“Based on the confirmed evidences, different responses of different prostate cancer cells can be explained.” We believe that the Discussion section has been improved sufficiently.

Reviewer 3 Report

The study is interesting and involve lot of work, using modren tools. The scientific interest is not very high, since it was already proved that DON is cytotoxic, not only for tumoral cells but also for cells involved in immune response

The article should be revised by a native English speaker (it contains many grammar mistakes)

The authors should follow carefully the Instructions for authors

Eg

Ref 16 is written in capitals letters. Ref 26 does not provide the title of the article

Author Response

Dear Reviewer,

I would like to submit the revised manuscript „Deoxynivalenol modulates the viability, ROS production and apoptosis in prostate cancer cells” for possible publication in Toxins. All the changes made in the manuscript has been made in the “Track changes” option in word with the exception of Figures and Tables (exchanged for corrected). In Figure 1 the one of diagrams has been doubled accidentally, but the result section was correct, thus it was changed for the correct one. According to the suggestion of The Associated Editor the manuscript has been corrected to avoid the self-plagiarism, nevertheless the names of the cell lines, media, kits, a commonly used phrases like “play a crucial role” or “increase in the number of apoptotic cells” etc. has not been changed due to the fact that they are just a long term names or phrases and cannot be rephrased, unfortunately. In other places, the sentences were rephrased.  The manuscript has been revised by a native speaker. Please, find the answers to Reviewers comments below, in italics.

“The study is interesting and involve lot of work, using modern tools. The scientific interest is not very high, since it was already proved that DON is cytotoxic, not only for tumoral cells but also for cells involved in immune response, The article should be revised by a native English speaker (it contains many grammar mistakes)” It is already known that DON is cytotoxic for many types of cells, but there is only a few studies documented that DON might be cytotoxic for human cancer cells and no research for the effect of DON in prostate cancer cells. The article has been revised by a native speaker at Foreign Language Teaching Centre at University.

“The authors should follow carefully the Instructions for authors Eg Ref 16 is written in capitals letters. Ref 26 does not provide the title of the article”  The instruction to authors has been carefully revised and references have been corrected.